# Preparation of $ZrO_2$/$TiO_2$/$Al_2O_3$ Nanofiltration Lab-Scale Membrane for Filtering Heavy Metal Ions

Jie Yang [1], Jinquan Sun [1,*], Huanzhong Bao [2], Song Li [2], Lianbao Zhang [2], Xinyue Bao [1], Fujie Li [1], Qingkun He [1], Na Wei [1], Kun Xie [1] and Wensheng Li [1]

[1] College of Material Science and Technology, Shandong University of Science and Technology, Qingdao 266510, China
[2] Zibo Megavision Membrane Environmental Protection Technology Co., Ltd., Zibo 255400, China
* Correspondence: skd992031@sdust.edu.cn

**Abstract:** $ZrO_2$ is an excellent ceramic preparation material that can maintain chemical stability in medium–strong acid and alkali environments. The sintering impregnation method was used to prepare the $ZrO_2$/$TiO_2$/$Al_2O_3$ composite nanofiltration membrane (ZTA membrane). Nano-$ZrO_2$, submicron $TiO_2$, and microporous $Al_2O_3$ were used as the surface layer, the transition layer, and the support layer, respectively. The structure and phase of the membrane were measured by scanning electron microscopy (SEM) and X-ray diffractometer (XRD). The composite membrane's retention, hydrophilic and hydrophobic properties were characterized and evaluated using a UV–Vis spectrophotometer, a water contact angle tester (WCA), and a dead-end filtration device. With the increase in separation layer deposition time, the retention rate of methyl blue increased, and the water flux decreased. At a deposition time of 75 min, the retention rate of methyl blue was more than 80%, and the water flux reached 337.5 $L \cdot m^{-2} \, h^{-1} \, bar^{-1}$ at $-1$ bar transmembrane pressure. The membranes are hydrophilic and have different interception abilities for metal ions, and the order of retention effect is $Ag^+ > Cu^{2+} > Mg^{2+} > Na^+$, and $Ag^+$ and $Cu^{2+}$ reached 65.3% and 50.5%, respectively. The prepared ZTA composite nanofiltration membrane has potential application value in heavy metal ion filtration.

**Keywords:** $ZrO_2$/$TiO_2$/$Al_2O_3$; nanofiltration; heavy metal ion interception



## 1. Introduction

In the current economic development, the demand for industrial water is increasing, and improper industrial emissions easily lead to environmental pollution and aggravate the shortage of water resources. Various countries have devoted themselves to treating and recycling industrial wastewater in recent years, and the environment has improved [1–4]. However, domestic water shortage because of poor water quality caused by heavy metal pollution is still worthy of attention [5,6]. By the end of 2020, 1/4 of the world's population still lacks safe drinking water services, and 489 million people still drink from untreated water sources [7]. Millions of children die yearly from acute diarrhea, and water pollution is a major contributor to such foodborne diseases [8]. Therefore, the treatment method for water pollution is still worth exploring. Sewage treatment methods mainly include physical, chemical and biological methods. According to different specific operations, they can be divided into centrifugation [9], adsorption [10], membrane separation [11], chemical precipitation [12], electrolysis [13], anaerobic biological treatment [14], etc. Compared with other traditional separation and extraction technologies, membrane technology stimulates the development of water treatment technology with its more convenient operation process and economical and environmentally friendly material storage. In recent decades, the desalination and purification of water, petrochemical, light industry, textile, food, biotechnology, medicine, environmental protection, and other fields have been widely developed [11,15].

Membrane separation technology relies on pore size screening and charge effects for the selective filtration of materials [16,17]. Membrane technology originated at the beginning of the 20th century and simultaneously has separation, concentration, purification, and refining functions. It can carry out efficient and energy-saving filtration at the molecular level, and the separation process is simple and easy to control, which has enormous social and economic benefits [18]. Among them, nanofiltration has been considered a promising membrane separation technology, which can intercept substances with molecular weights ranging from 200 to 2000 Da [19,20]. Some polyvalent ions can also be trapped in special nanofiltration membranes due to the screening of pore size and the presence of surface charges. The ion capture property of the membrane material is beneficial to the filtration and treatment of heavy metal ions in wastewater and the reduction of heavy metal pollution [21–23]. With the development of society and the growth of water resource utilization, the demand for membrane materials is also increasing. The application of the organic membrane is limited because it is easily polluted and is not resistant to acids and alkalis [15–18]. While organic materials are modified to improve their properties, inorganic materials have been paid more and more attention due to their acid and alkali resistance, high-temperature resistance, and high chemical stability [20,24–26]. The primary inorganic membrane materials include titanium oxide ($TiO_2$), zirconia ($ZrO_2$) and alumina ($Al_2O_3$), etc. [27–30].

As a form of membrane preparation, an asymmetric membrane usually consists of two parts, one is a selection layer responsible for separation performance, and the other is a porous support layer responsible for providing mechanical strength [31–33]. The structure can reduce the filtration resistance of membrane materials, increase the permeability of the membrane, and provide the necessary mechanical strength, which is a typical format for membrane material modification [21,34,35]. Tao Meng et al. [36] deposited nano-$SiO_2$ particles on the surface of the support as a grafting platform and further modified the membrane with hydrophilicity–hydrophobicity. Xiquan Cheng et al. [37] designed selective asymmetric composite membranes that significantly reduced the mass transfer resistance and prepared composite membranes that could exhibit ultrafast permeance.

$ZrO_2$ is an excellent ceramic preparation material that can maintain its chemical stability in medium–strong acid and alkali environments [27]. In this paper, the asymmetric composite film was prepared by the hydrothermal method using nano-zirconia as the separation material, and the surface layer of $ZrO_2$ was modified by adding polyethylene imide to give it the ability of ion interception [38]. In the process of asymmetric membrane preparation, a reasonable pore size difference is an essential factor affecting the effect of membrane preparation. If the pore size difference between the carrier and the microporous layer is too significant, the flatness of the prepared separation layer will be seriously affected. One method is to increase the viscosity or tightness of the film by adding an auxiliary agent [26]. Another way is to prepare multilayer materials by adding transition layers to reduce the pore size difference and improve the smoothness of the separation layers [39,40]. Previous researchers have done a lot of work on the former and less on the latter. To obtain a continuous dense membrane material and ensure reasonable pore size difference transition, submicron particles were added to prepare the intermediate layer. Submicron $TiO_2$ material was used in the middle layer. Titanium dioxide is a high-quality ceramic material with the characteristics of easy access and low cost. Compared with the deformation and cracking of $ZrO_2$ during heating, the thermal expansion coefficient of $TiO_2$ has little change, and the film formation effect is better, which can improve the success rate of film production [26,41]. It is a hydrophilic material and can enhance the flux of the membrane materials [29]. $ZrO_2/TiO_2/Al_2O_3$ composite nanofiltration membranes (ZTA membranes) were prepared on an $Al_2O_3$ support with a pore size of 1–3 μm by sintering impregnation.

## 2. Experimental

### 2.1. Materials

The reagents and materials used in the experiment are shown in Table 1, all of which have been used.

**Table 1.** The experimental reagents and materials used in the experiments.

| Materials | Chemical Formula | Manufacturer/Information |
|---|---|---|
| Alumina | $\alpha$-Al$_2$O$_3$ | $\Phi$140 $\times$ 3 mm, aperture size ranges from 1 to 3 $\mu$m |
| Titanium dioxide (100 nm–600 nm) | TiO$_2$ | Jinan Yuxing Chemical Co., Ltd. (China) |
| Methylene blue | C$_{16}$H$_{18}$N$_3$ClS | Shandong West Asia Chemical Industry Co., Ltd. (China) |
| Methyl blue | CHN$_3$Na$_2$O$_9$S$_3$ | Sinopharm Chemical Reagent Co., Ltd. (China) |
| Ammonium hydroxide | NH$_3$·H$_2$O | Sinopharm Chemical Reagent Co., Ltd. (China) |
| Polyvinyl alcohol | PVA (degree of polymerization: 1750 $\pm$ 50) | Sinopharm Chemical Reagent Co., Ltd. (China) |
| NaCl | NaCl | Sinopharm Chemical Reagent Co., Ltd. (China) |
| AgNO$_3$ | AgNO$_3$ | Sinopharm Chemical Reagent Co., Ltd. (China) |
| CuSO$_4$ | CuSO$_4$ | Sinopharm Chemical Reagent Co., Ltd. (China) |
| Polyethylene imine | PEI (Mw = 70,000 Da, 50% aqueous solution) | Shanghai Macklin Biochemical Co., Ltd. (China) |
| Zirconium nitrate pentahydrate | Zr(NO$_3$)$_4$·5H$_2$O | Shanghai Macklin Biochemical Co., Ltd. (China) |
| Glycerine | C$_3$H$_8$O$_3$ | Tianjin Guangfu Science and Technology Development Co., Ltd. (China) |

### 2.2. Membrane Preparation

#### 2.2.1. Preparation of the Intermediate Layer

The intermediate layer was prepared using a dip-coating and sintering procedure. The suspension of titanium dioxide was prepared by ultrasound, in which 10 wt% PVA solution was used as a solvent, and the content of solid particles was 10 wt%. A 1.5–2 mL suspension of TiO$_2$-PVA was absorbed with a tip glue dropper and uniformly coated on the surface of the $\alpha$-Al$_2$O$_3$ carrier. The obtained samples were dried at room temperature. After complete drying, the above steps were repeated once again, and the obtained sample was heated to 800 °C at a heating rate of 8 °C/min for calcination for 2 h to obtain the TiO$_2$–Al$_2$O$_3$ composite membrane. A group of control experiments without PVA was established. TiO$_2$ was dispersed with deionized water to prepare a 10% solid coating solution. The other steps were the same as the membrane preparation process with the PVA addition.

#### 2.2.2. Preparation of ZTA Composite Membrane

Nano-zirconia was synthesized by the laboratory hydrothermal method. Ammonium hydroxide was used as a precipitant, and water and glycerine as the medium. A 25 w% ammonium hydroxide solution was slowly added to 1 mol/L zirconium salt solution. This process involved the continuous agitation of the solution. The pH of the solution was observed and adjusted to about pH 10 with a pH meter, then 1 w% glycerin was added to it and stirred evenly until a milky white uniform suspension formed. The obtained precursor was added to a 100 mL high-pressure digestion tank and filled to 70%~80% with deionized water. After complete mixing, the precursor was reacted at 180 °C for 12 h and cooled to room temperature. The precursor was then washed, dried and milled for reserve use.

The suspension of ZrO$_2$ was prepared by ultrasound, in which a PEI solution of 5 mg·mL$^{-1}$ was used as a solvent, and the content of solid particles was 100 mg/mL. The TiO$_2$–Al$_2$O$_3$ composite membrane was placed on the end filtration device (Figure 1), sealed, and connected with a vacuum pump. The suspension was then dropped onto the material surface with a rubber head dropper to keep TiO$_2$–Al$_2$O$_3$ soaked. During the preparation process, the transmembrane pressure was −1 bar, and the PEI–ZrO$_2$ separation membrane was stacked on the TiO$_2$–Al$_2$O$_3$ surface. The thickness of the separation layer could be adjusted by the concentration of PEI–ZrO$_2$ or the preparation time. The ZTA composite film was dried at room temperature (20 °C) for 48 h for subsequent testing. The same

procedure was repeated to prepare a group of ZTA composite membranes without PEI as a comparison.

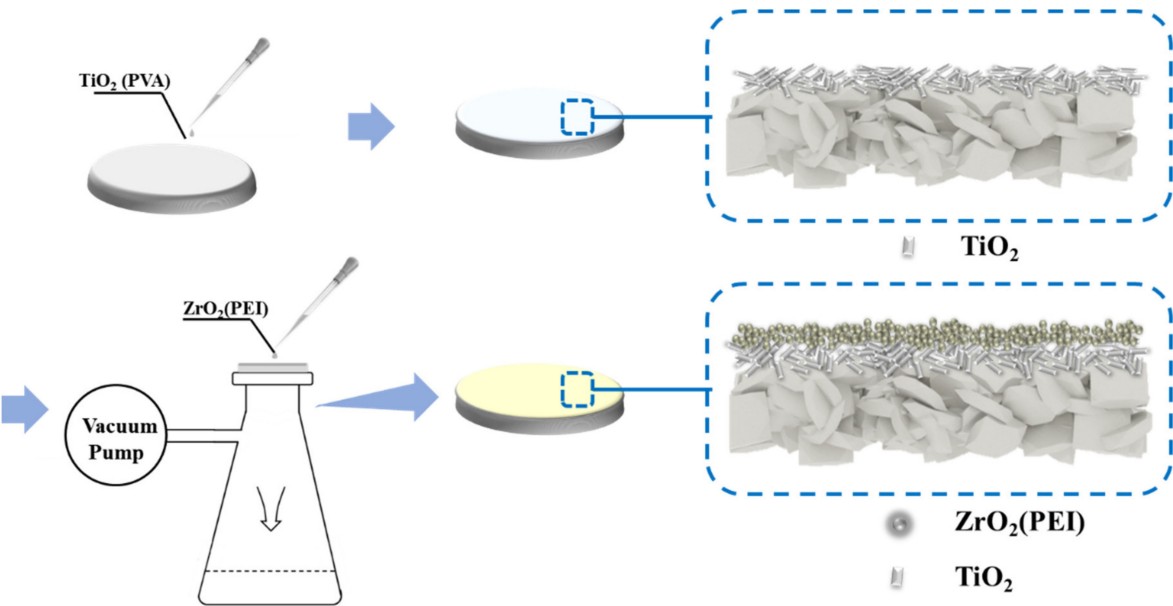

**Figure 1.** Schematic diagram of the $ZrO_2/TiO_2/Al_2O_3$ composite membrane preparation process.

### 2.3. Characterization

The microstructure of the $ZrO_2$ composite membranes was characterized by a field emission scanning electron microscope (FESEM, Nova Nano SEM 450, FEI company, Hillsboro, OR, USA). The phase compositions were analyzed by X-ray diffraction (XRD, Rigaku D/max 2200 X-ray diffractometer, Nippon Electronics Co (JEOL), Shojima City, Tokyo, Japan) at 40 kV and 40 mA with a scan step of 8° (Cu Kα radiation, λ = 0.154 nm). Water contact angles (WCA) were measured using a contact angle measuring instrument (DSA30, Kruss company, Hamburg, Germany). A circulating water vacuum pump (SHZ-D(iii), Buncy Instrument Technology Co., Ltd., Shanghai, China) was used to measure the water flux of the membrane. Filtered liquids were characterized using a UV–Vis spectrophotometer (Agilent CARY 60, Agilent Technologies Co. Ltd., Chaoyang District, Beijing, China) to display the membrane separation performance visually. The ion concentration in the filtrate was quantitatively analyzed using a conductivity meter (DDS-11A, Rex Electric Chemical, Shanghai, China).

### 2.4. Nanofiltration Performance

Permeability was characterized using a laboratory suction filtration device (−1 bar). The water permeance of the composite membrane was calculated by:

$$Q = V/(A \times t \times \Delta P) \tag{1}$$

where $Q$ is the normalized water permeability (l $L \cdot m^{-2} h^{-1} bar^{-1}$), $V$ is the filter flow (L), $A$ is the effective membrane area ($m^2$), $t$ is the experiment time (h), and $\Delta P$ is the transmembrane pressure (bar). The rejection was calculated by:

$$R = \left(1 - \frac{C_p}{C_f}\right) \times 100\% \tag{2}$$

where $C_f$ and $C_p$ are the solute concentrations in the feed and permeate, respectively, and the attention of salts was measured using a conductivity meter.

### 3. Results and Discussion

*3.1. Characterization of ZrO$_2$*

Figure 2 shows the microstructure and phase composition of nano-ZrO$_2$ particles synthesized by the hydrothermal method. It can be seen that the hydrothermal-synthesized ZrO$_2$ presents uniform spherical particles (Figure 2a,b), and the acquisition of uniform particles is conducive to the preparation of a membrane surface with uniform osmotic pores. The ZrO$_2$ particle size is mainly distributed in the 20 nm to 40 nm range (Figure 2c), which is conducive to forming small aperture surfaces. After hydrothermal synthesis, ZrO$_2$ was analyzed by XRD phase, and the results are shown in Figure 2d. After calibration, the corresponding standard card was 49-1642. The prominent characteristic peaks $2\theta = 30.119°$, $34.959°$, $50.219°$, and $59.738°$ correspond to the (111), (200), (220), and (311) crystal planes of ZrO$_2$, respectively, and the other peaks also corresponded to the crystal planes specified in the card. Therefore, it can be considered that the prepared ZrO$_2$ is in a pure uniform phase. The homogeneity of the single pure phase is beneficial to reduce the negative effect of the material itself in the process of membrane production and to improve the quality of membrane production.

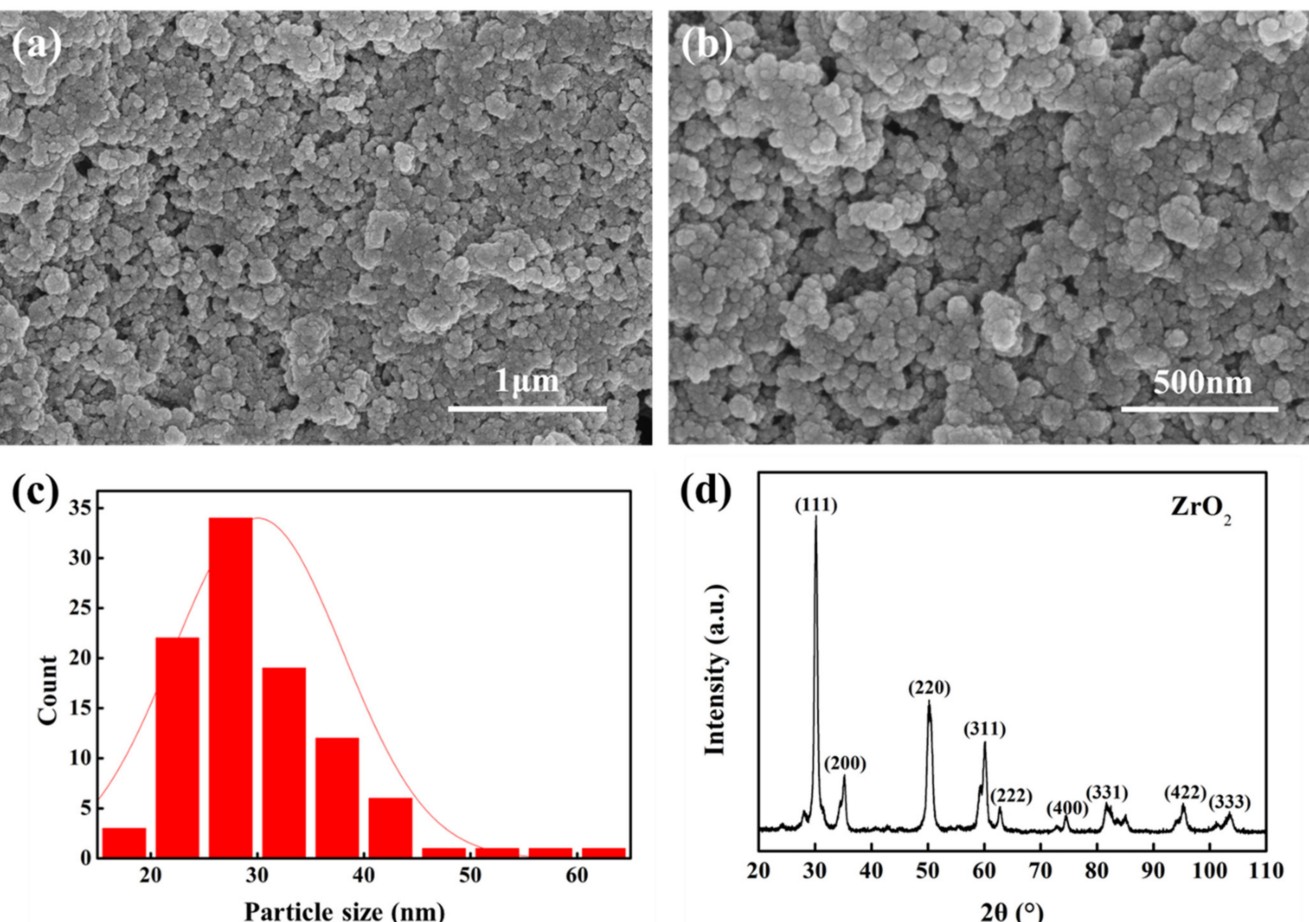

**Figure 2.** ZrO$_2$ particles prepared by the hydrothermal method (**a**,**b**), particle size statistics (**c**), and X-ray phase analysis (**d**).

*3.2. Characterization of Support*

Figure 3a shows the surface topography of the Al$_2$O$_3$ support. The aperture of the support body is at the micron level. An in situ hydrothermal reaction was used to synthesize the ZrO$_2$ membrane as a control. The in situ hydrothermal reaction is used to place the support directly in the reactor, and zirconia is directly generated and deposited on the surface of the carrier. Figure 3b,c show the microscopic morphology after the reaction.

Due to the significant pore size difference between the hydrothermal-synthesized $ZrO_2$ nanoparticles and the alumina support body, no dense and continuous surface structure is formed, which is not conducive to the deposition and molding of the membrane. There are two ways to improve this phenomenon. One is to increase the viscosity of the surface coating liquid to enhance the surface material's supporting force and to make the surface film material uniformly loaded on the surface of the support body. Secondly, it can be improved by adding a transition layer. A transition layer of appropriate size between the nano-surface layer and the microporous support body can fill the pores of the support body and support the surface material simultaneously to obtain a continuous and dense surface structure. Therefore, in this paper, an intermediate layer was added between the alumina support and zirconia separation layer to improve the preparation performance of the membrane.

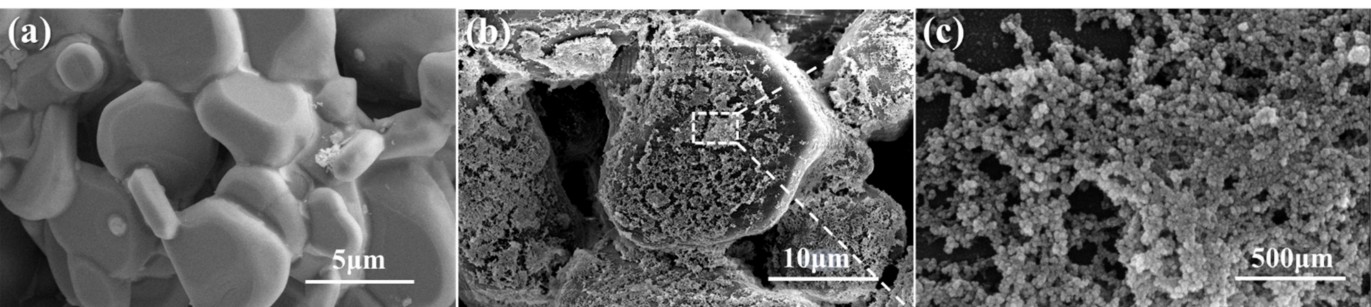

**Figure 3.** Surface morphology of the $Al_2O_3$ support (**a**), the microstructure of the $ZrO_2$ membrane prepared using the in situ hydrothermal reaction on the support (**b**,**c**).

*3.3. Characterization of ZTA Composite Membranes*

3.3.1. Characterization of Intermediate Layer

Due to the hydrophilic and bacteriostatic properties of $TiO_2$, it is beneficial to improve the permeability and durability of the membrane [41]. $TiO_2$ is selected as the intermediate layer material and prepared on the support to form the transition between the surface layer and the support to reduce the pore size difference. It is convenient to prepare a continuous dense $ZrO_2$ separation layer structure. Figure 4a,b are SEM morphologies of the $TiO_2$ intermediate layer in which $TiO_2$ particles are distributed in a short rod shape. PVA was not added in Figure 4a but in Figure 4b. By comparing the morphology of the matrix (Figure 3a), it can be seen that the prepared $TiO_2$ intermediate layer has effectively reduced and filled the support pores. By comparing Figure 4a,b, it can be seen that the addition of PVA makes $TiO_2$ more evenly dispersed and forms a smoother intermediate layer on the surface of the support. This is because the addition of PVA can increase the viscosity of the membrane-forming liquid, improve the surface's continuity in the film preparation process, and obtain a smooth and continuous membrane [42].

At the same time, $TiO_2$ with good solubility in water can obtain good dispersion in the PVA solution, which is conducive to preparing the coating solution. After subsequent characterization, the $TiO_2$ intermediate layer composed in the experiment has higher permeability and lower retention performance. The flux is about 3000 $L \cdot m^{-2} \, h^{-1} \, bar^{-1}$, and the rejection rate is almost 0%. The XRD phase analysis of the surface of the $TiO_2$–$Al_2O_3$ film after the preparation of the middle layer is shown in Figure 4c. By comparing and analyzing the PDF card, the calibration phase is single $TiO_2$. That is, the middle layer is a stable $TiO_2$ phase. According to the statistical analysis of the material particle size, the diameter range of the $TiO_2$ is mainly between 200–500 nm (Figure 4d).

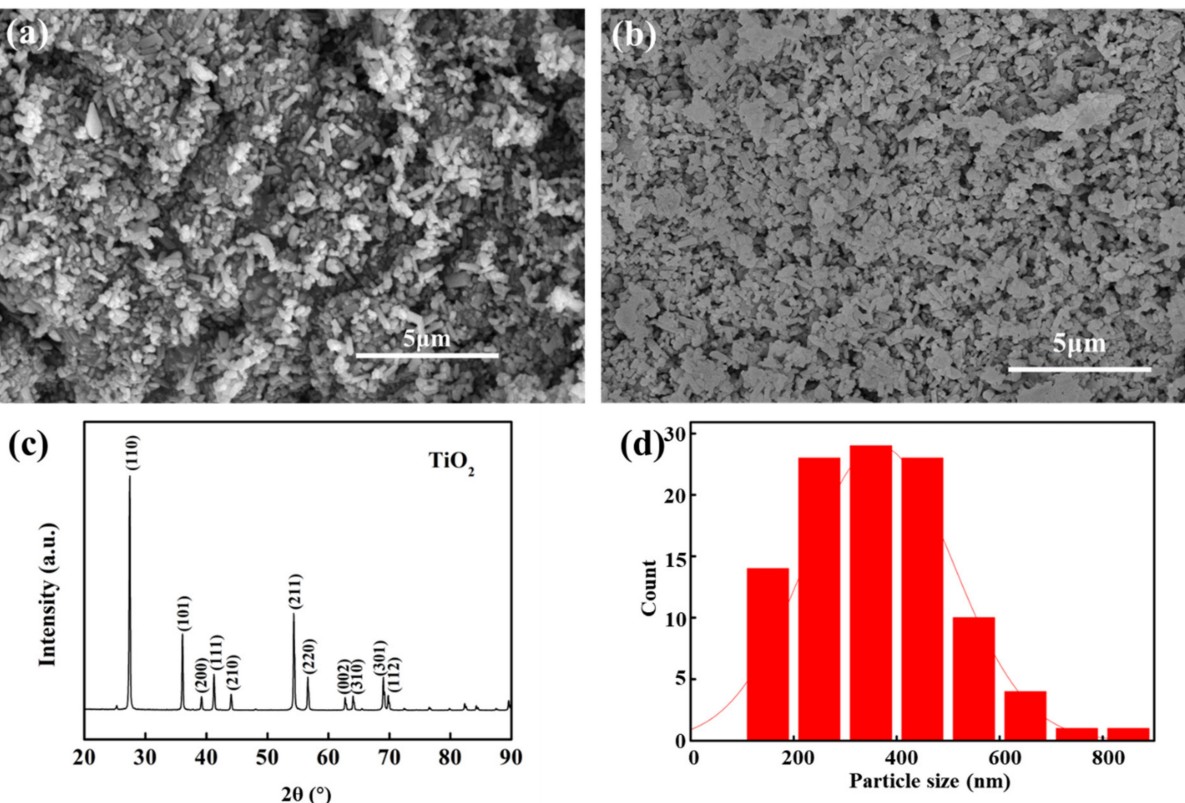

**Figure 4.** The morphology of the TiO$_2$–Al$_2$O$_3$ membrane surface ((**a**) without adding PVA, (**b**) with adding PVA), XRD analysis of TiO$_2$–Al$_2$O$_3$ membrane surface after the preparation of the intermediate layer (**c**) and particle size statistics of the TiO$_2$ powder (**d**).

### 3.3.2. Morphology and Phase of ZTA Composite Membrane

The zirconia surface layer was prepared by deposition through a terminal filtration device. Figure 5 shows the surface and section microstructure characterization of the composite film and surface XRD phase analysis. To improve the membrane's interception performance and prepare a ZTA composite membrane with better and more stable performance, PEI was introduced to prepare the surface layer in this experiment. PEI has high cationic and hydrophilic properties, which can affect the hydrophilicity of the membrane and ion filtration [43]. Figure 5 shows the microstructure characterization of the composite film and surface XRD phase analysis. The surface morphology of the final prepared ZTA composite membrane is shown in Figure 5a. It can be seen from Figure 5a that the surface of the composite film is smooth and flat, and the particles are evenly and densely packed. This morphology can provide good structural support for the performance of the composite membrane. The morphological characterization of the cross-section of the composite membrane is shown in Figure 5b. The thickness of the intermediate layer and the separation layer is 28.4 μm. It can be seen from the cross-sectional morphology that the separation layer of the ZTA composite membrane is tightly combined with the matrix. Some of the particles filled in the particle gap on the surface of the support body, forming a serrated bond surface, increasing the bond area and improving the bond strength between the membrane and the support.

From the XRD pattern (Figure 5c), it can be concluded that a single ZrO$_2$ separation layer was distributed on the surface of the ZTA composite membrane. Compared with the standard card, there was no change in the distribution of detected characteristic peaks, so it can be considered that the PEI had no structural influence on the phase changes of ZrO$_2$. PEI can make particles evenly arranged in the separation layer and promote the formation of an ordered ZrO$_2$ separation layer.

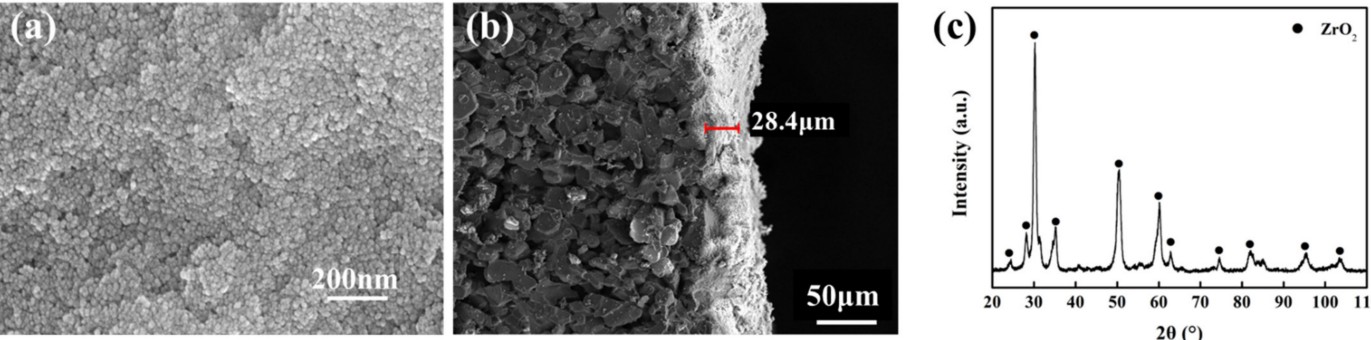

**Figure 5.** Microscopic morphology of the $ZrO_2/TiO_2/Al_2O_3$ composite nanofiltration membrane (ZTA composite membrane) (The surface (**a**) and the cross-section (**b**) of the composite membrane). (**c**) X-ray phase analysis of the ZTA composite membrane surface.

### 3.3.3. Filtration Performance of the ZTA Composite Membrane

Hydrophilicity is a crucial factor affecting the performance of composite membranes. In this paper, 5 μL of deionized water was dropped on the surface of the membrane to test the water contact angle. The water contact angle was tested on the $TiO_2$ intermediate layer, the surface of the ZTA composite membrane without PEI addition and the surface of the ZTA composite membrane with PEI addition. The results are shown in Figure 6. It can be seen from Figure 6a that the contact angle of the intermediate layer was 38.2°, which belongs to the hydrophilic category. The hydrophilicity of the $ZrO_2$ surface prepared with added glycerol (Figure 6b) was lower than that of the hydrophilic $TiO_2$ material (Figure 6c). The results show that after introducing the PEI-modified $ZrO_2$ separation layer, the hydrophilicity was improved, and the contact angle was at 20.5°. This was mainly due to the large number of hydrophilic groups in PEI, resulting in the hydrophilic modification of the separation layer.

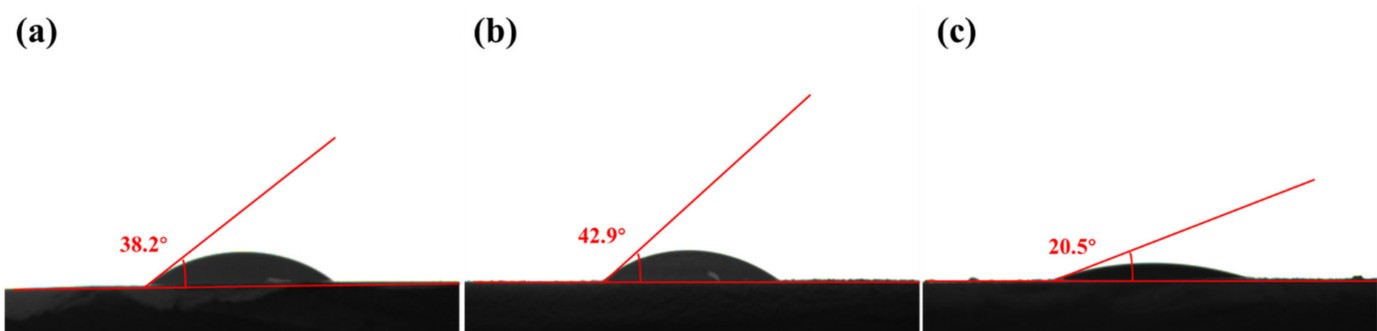

**Figure 6.** Water contact angle measurement result ((**a**) $TiO_2$–$Al_2O_3$ membrane surface, (**b**) ZTA composite membrane surface without PEI addition, and (**c**) ZTA composite membrane surface).

We conducted pure water flux tests on the membranes with different surface layers through dead-end filtration devices. The result is shown in Figure 7. It can be seen that the flux of the support without surface treatment was at a relatively high level, about 8000 $L \cdot m^{-2} \, h^{-1} \, bar^{-1}$. After preparing the $TiO_2$ intermediate layer, the flux reduced to 3000 $L \cdot m^{-2} \, h^{-1} \, bar^{-1}$. After preparing the $ZrO_2$ separation layer, the change stabilized at 337.5 $L \cdot m^{-2} \, h^{-1} \, bar^{-1}$.

To evaluate the retention performance of the ZTA composite membrane, the experiment uses methyl blue as a calibrator. The results are shown in Figure 8a. It can be seen that as the number of separation layer depositions increases, the retention performance gradually increases. At the same time, it is accompanied by a constant decrease in flux. Aperture blocking and membrane thickening lead to decreased flux and improved interception performance.

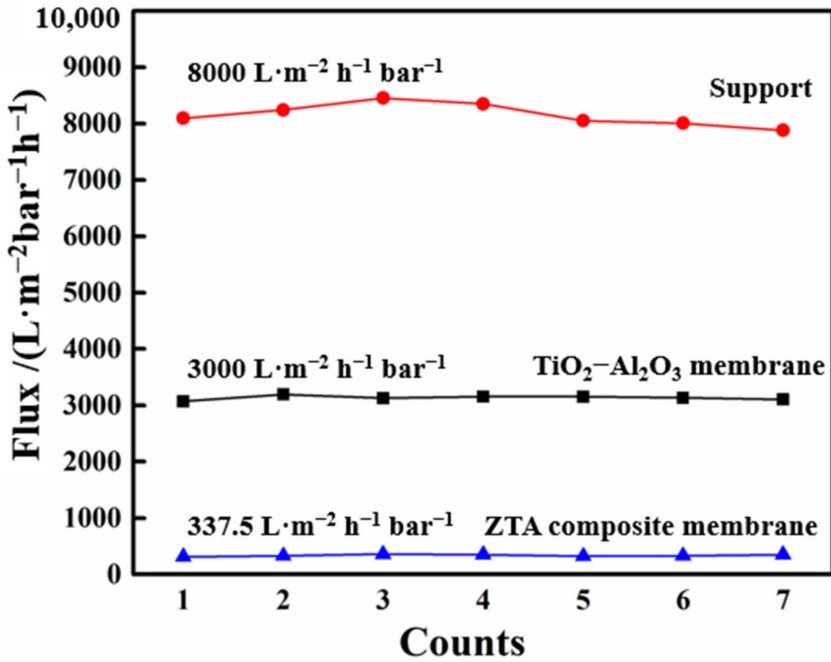

**Figure 7.** Pure water flux evaluation of support, $TiO_2$–$Al_2O_3$ membrane and ZTA composite membrane.

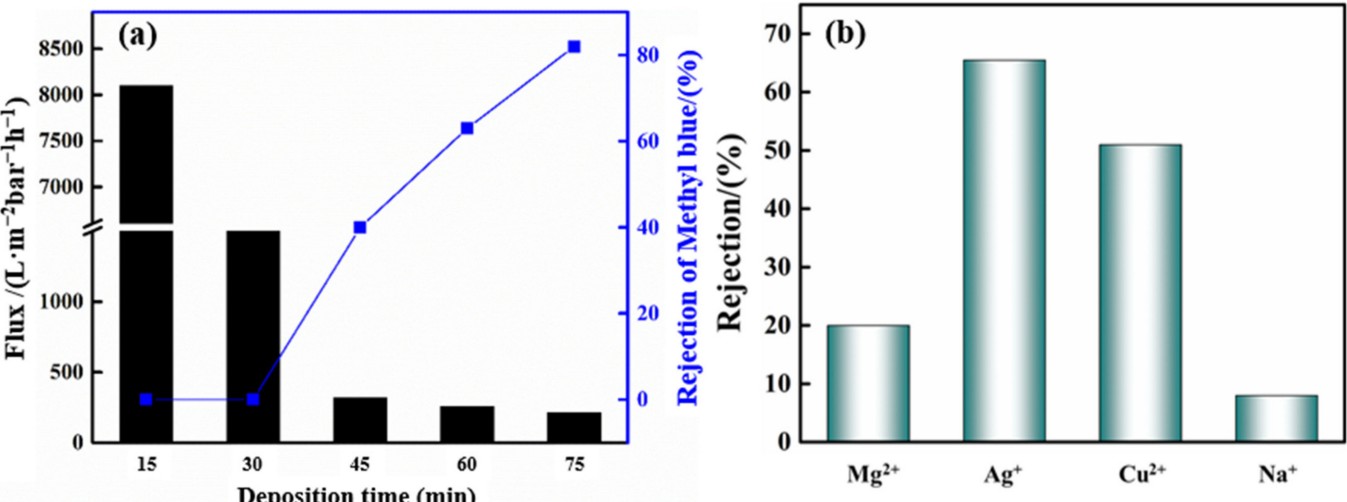

**Figure 8.** Effect of $ZrO_2$ surface layer deposition time on pure water flux and methyl blue interception of the ZTA composite membrane (**a**), $Ag^+$, $Cu^{2+}$, $Mg^{2+}$, $Na^+$ cationic interception effect of ZTA membrane after 75 min deposition (**b**).

The retention effect of the ZTA composite membrane on cations in different inorganic salt solutions is shown in Figure 8b. The order of the retention effect of the cationics is $Ag^+$ > $Cu^{2+}$ > $Mg^{2+}$ > $Na^+$. This trend indicates that the surface of the composite membrane is positively charged, due to the addition of PEI in the surface layer, which increases the cationic charge density of the surface layer and affects the cation filtration [44]. Driven by transmembrane pressure, water molecules can penetrate through the pores of the separation layer [45]. Under the action of pore size sieving and chelation adsorption, the charged ions are selectively trapped by the separation layer. The adsorption effect depends on the chemical potential of the ion. The results show that the interception effect of $Ag^+$ and $Cu^{2+}$ is better than that of $Mg^{2+}$ and $Na^+$, which fits the chemical potential order $Ag^+$ (+0.7996 V) > $Cu^{2+}$ (+0.340 V) > $Mg^{2+}$ (−2.372 V) > $Na^+$ (−2.71 V). PEI molecules are adsorbed on $ZrO_2$

particles through electrostatic and hydrogen bonding to increase the charge density on the membrane surface [46]. Coordination reactions occur between -NH$_2$ functional groups and heavy metal ions, such as Ag$^+$ and Cu$^{2+}$, forming coordinated covalent bonds to form chelates, thus producing ion adsorption and retention. The principle is shown in Figure 9. In conclusion, the ZTA composite membrane has potential application value in retaining heavy metal ions.

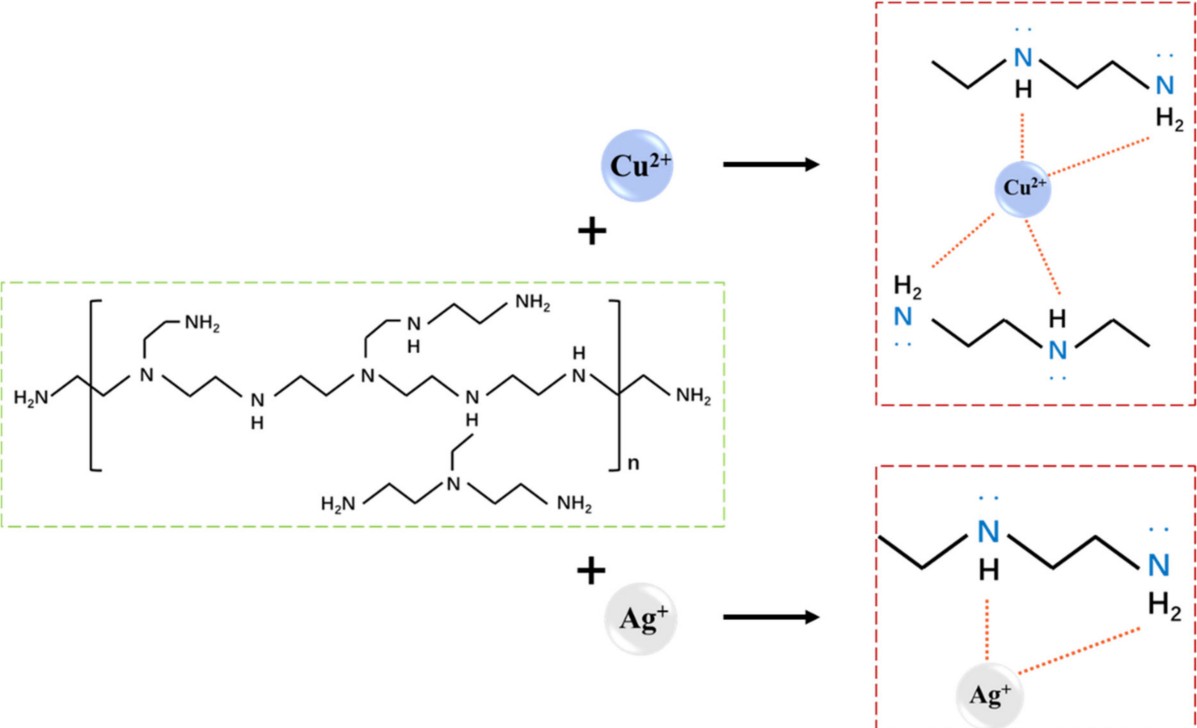

**Figure 9.** Schematic diagram of the molecular structure of PEI and its coordination process with Cu$^{2+}$ and Ag$^+$.

## 4. Conclusions

In this study, we constructed a ZrO$_2$/TiO$_2$/Al$_2$O$_3$ composite nanofiltration membrane (ZTA composite membrane), which used nano-ZrO$_2$ as the surface layer, submicron TiO$_2$ as the transition layer, and microporous Al$_2$O$_3$ as the support layer. TiO$_2$ was used to prepare the intermediate layer, which reduced the aperture difference between the support and the separation layer, and prepared a smooth and flat separation layer surface. The addition of PVA makes the intermediate layer more uniform and flatter. PEI makes the separation layer show a smooth and compact microstructure, improving the composite membrane's preparation effect and hydrophilicity. The ZTA composite membrane prepared in the laboratory has a specific ion interception ability. Although the surface bonding strength of the ZTA composite membrane prepared by this method needs to be improved, it has particular application potential in heavy metal ion filtration. To further explore the application potential and practical value of the ZTA composite membrane, further studies will be carried out on the interpretation of the influence of the optimized preparation parameters and properties of the material.

**Author Contributions:** Conceptualization, J.S. and K.X.; methodology, N.W.; software, J.Y. and F.L.; validation, S.L. and L.Z.; formal analysis, J.Y.; investigation, J.Y.; resources, J.S.; data curation, J.Y.; writing—original draft preparation, J.Y.; writing—review and editing, J.Y.; visualization, X.B. and W.L. supervision, Q.H.; project administration, J.S.; funding acquisition, H.B. All authors have read and agreed to the published version of the manuscript.

**Funding:** The authors would like to acknowledge the support from the MABR membrane equipment industrialization project (2019JZZY020224) and the equipment support from the Shandong University of Science and Technology (College of Material Science and Technology).

**Institutional Review Board Statement:** Not applicable.

**Informed Consent Statement:** Informed consent was obtained from all subjects involved in the study.

**Data Availability Statement:** All research results are presented in the paper, and no other supplementary materials are available.

**Acknowledgments:** The authors gratefully acknowledge the support provided by Jinquan Sun and Qi Yan for their guidance in the experiment and writing. All authors were informed and consented to publication.

**Conflicts of Interest:** The authors declare no conflict of interest.

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
