# Peer review of "Preparation of ZrO2/TiO2/Al2O3 Nanofiltration Lab-Scale Membrane for Filtering Heavy Metal Ions"

_coatings, doi:10.3390/coatings12111681_

Round 1
Reviewer 1 Report
Dear Editor
With Greeting,
Thank you for submitting your article to the Coatings (ISSN 2079-6412). I have gone through the manuscript; the paper entitled “Preparation of ZrO2/TiO2/Al2O3 Nanofiltration Lab Scale Membrane for Filtering Heavy Metal Ions” explains cost effective and easy route to face out the impurity of hevay metal. The authors had successfully instigated the impact of composite.
This is suitable for Coatings with some minor revisions:
1. The part 1. Introduction is little short, kindly elaborate this section with some recent publications.
2. Here, I would like to suggest kindly add EDX spectroscopy for better understanding of comospite.
3. Also disucss about its thermal stability.
With these corrections, this manuscript is suitable for Coatings Journal.
With Regards
Sincerely
Author Response
1. The part 1. Introduction is little short, kindly elaborate this section with some recent publications.
A: Through literature review, the selection of ZrO2 and TiO2 in this paper is further supplemented, and corresponding references [39-43] are added.
2. Here, I would like to suggest kindly add EDX spectroscopy for better understanding of comospite.
A: Due to the problem of experiment allocation, we did not conduct EDX experiment, but we will continue to pay attention to this part of the analysis in the future.
3. Also disucss about its thermal stability.
A: Since PEI polymer was used to modify the material in the surface preparation process, the present material properties are suitable for maintaining at low temperature (below 200℃). The improvement of ion interception after high temperature treatment of the composite film will be explored in the subsequent research.
Reviewer 2 Report
The authors report the preparation of ZrO2/TiO2/Al2O3 composite nanofiltration membranes. The manuscript flows well and is readbale.
The introduction is written in a general way and does not cover the important literature regarding ZrO2 and TiO2 membranes. The manuscript lacks novelty. Literature review shows that this filed is very-well investigated.
1- In line 149 the authors have written that “The ZrO2 particle size used in the experiment is mainly distributed in the range of 149 40 nm to 60nm (Figure 2(c))”. However, Fig 2c shows that the size of NPs are mainly 20-40 nm. This needs to be modified accordignly.
2- In line 206 “Fig 2d”should change to “Fig 4d”.
3- 3.3. and 3.3.2 are the same. Change is required accordingly.
4- In Fig. 5c is related to the ZTA membrane, not ZrO2.
5- In line 291, “a kind of”should be deleted.
6- The SEM cross-section image (5c) is not clear. Improvemnet is advised.
Author Response
The introduction is written in a general way and does not cover the important literature regarding ZrO2 and TiO2 membranes.
A: Through literature review, the selection of ZrO2 and TiO2 in this paper is further supplemented, and corresponding references [39-43] are added.
The manuscript lacks novelty. Literature review shows that this filed is very-well investigated.
A: (Innovation) In the process of asymmetric membrane preparation, a reasonable pore size difference is an essential factor affecting the effect of membrane preparation. If the pore size difference between the carrier and the microporous layer is too significant, the flatness of the prepared separation layer will be seriously affected. One method is to increase the viscosity or tightness of the film by adding an auxiliary agent. Another way is to prepare multilayer materials by adding transition layers to reduce the pore size difference and improve the smoothness of the separation layers. Previous researchers have done a lot of work on the former and less on the latter. To obtain continuous dense membrane material and ensure reasonable pore size difference transition, submicron particles were added to prepare the intermediate layer in this paper. The innovation of this paper is the preparation of multilayer composite membranes capable of ion interception. This structure has the potential to improve the ion retention performance and increase the flux.
1. In line 149 the authors have written that “The ZrO2 particle size used in the experiment is mainly distributed in the range of 149 40 nm to 60nm (Figure 2(c))”. However, Fig 2c shows that the size of NPs are mainly 20-40 nm. This needs to be modified accordignly.
A: Thank you for pointing out, We have checked the original photo data of the test, and the description of particle size in the paper is not rigorous, which has been changed.
2. In line 206 “Fig 2d”should change to “Fig 4d”.
A: After inspection, the wrong marks have been corrected.
3. 3.3. and 3.3.2 are the same. Change is required accordingly.
A: 3.3.2 has been modified to “Morphology and phase of ZTA composite membrane”.
4. In Fig. 5c is related to the ZTA membrane, not ZrO2.
A: Fig. 5 of the ZTA membrane has been corrected and analyzed in detail.
5. In line 291, “a kind of”should be deleted.
A: "a kind of" has been removed.
6. The SEM cross-section image (5c) is not clear. Improvemnet is advised.
A: The SEM cross section annotation of Fig. 5(b) has been enlarged.
Reviewer 3 Report
In the presented manuscript authors prepared ZrO2/TiO2/Al2O3 nanofiltration membrane (ZTA membrane) by sintering impregnation method. The idea is very interesting and can attract the scientific audience. The experiments are well-planned and obtained materials were thoroughly characterized. However, there are some changes authors should make before the paper can be finally accepted for publication in "Coatings" journal.
1. The English should be checked by native speaker. The language style in whole paper should be significantly improved.
2. Lines 33-35: Authors should put reference after this sentence “By 33 the end of 2020, 1/4 of the world’s population still lacks safe drinking water services, and 34 even 489 million people are still drinking untreated water sources. Millions of children”.
3. Lines 97-98: Please, correct the sentence “Immersed Al2O3 in the mixture and removed it after 1min”. It is very unclear.
4. Lines 108-109, Section 2.2.2: Please, specify which reaction kettle (autoclave) was used for hydrothermal synthesis in order to ensure better reproducibility of your experiments.
5. Lines 114-115: Authors said: “The TiO2-Al2O3 composite membrane was placed on the end filtration device (Figure 1(b)) …”. However, I didn’t see a or b marks in Figure 1. Authors should add that or clear the b mark from the text.
6. Lines 149-150: Authors said: “The ZrO2 particle size used in the experiment is mainly distributed in the range of 40 nm to 60nm (Figure 2(c))…”. I think it is incorrect since I saw another dominant size distribution (20-40 nm) from the mentioned figure. Also, remove “used in the experiment” from the text to make sentence more clear.
7. In my opinion the whole 3.2. Section should be moved to supplementary since it explained reasons for choosing to put intermediate layer (TiO2) between ZrO2 and matrix.
Lines 163-164: Authors said: “The pore size of the support body ranges from 500 nm to 3500 nm”. From obtained micrographs (Figure 3), I could not see that since it wasn’t measured.
It would be great if authors should provide BET analyses fore some samples to make some conclusions about specific surface area and porosity.
8. Lines 191-193: “By comparing the morphology of the matrix (Figure 3(c)), it can be seen that the prepared TiO2 intermediate...” I think it is a Figure 3(a).
9. Since you provided SEM micrographs of intermediate TiO2 layer with and without PVA (Figure 4), you should mention preparation procedures of TiO2 layer with and without PVA in experimental section.
10. Lines 212-213: “Zirconia surface layer was prepared by deposition through terminal filtration device. The microscopic morphology of the membrane is shown in Figure 5”. Figure 5(a)?
11. Lines 242-246, Section 3.3.3: The paragraph “The hydrophilicity of ZrO2 prepared with glycerol added was lower than that of TiO2 hydrophilic material. The results show that after the introduction of PEI-modified ZrO2 separation layer, the hydrophilicity was slightly improved, and the contact angle was stabilized at 14.1°. This is mainly due to the large number of hydrophilic groups in PEI, resulting in hydrophilic modification of the separation layer.” It is confusing since water contact angle measurement result was not provided for ZrO2 prepared with glycerol. Also, I did not see the comparison between ZrO2 prepared with glycerol and PEI-modified ZrO2 before (SEM, etc.).
12. Lines 298-299, Conclusions: Authors said: “The addition of PEI improved the retention and adsorption ability of the membrane surface”. However, there were not performed experiments which confirms this statement (etc. comparison of retention and adsorption ability of the membrane surface between ZrO2 and PEI-modified ZrO2).
Author Response
1. The English should be checked by native speaker. The language style in whole paper should be significantly improved.
A: The article has been professionally proofread, and some sentences have been revised, and the wording and grammar problems have been corrected in order to better meet the quality requirements of Coating.
2. Lines 33-35: Authors should put reference after this sentence “By 33 the end of 2020, 1/4 of the world’s population still lacks safe drinking water services, and 34 even 489 million people are still drinking untreated water sources. Millions of children”.
A: This is a great suggestion. It has been added and modified.
3. Lines 97-98: Please, correct the sentence “Immersed Al2O3 in the mixture and removed it after 1min”. It is very unclear.
A: This sentence has been redescribed.
There is a grammatical error in this sentence, which has been modified to “The 1.5-2 ml suspension of TiO2-PVA was absorbed with a glue tip dropper and uniformly coated on the surface of the α-Al2O3 carrier. The obtained samples were dried at room temperature. After complete drying, the above steps were repeated once, …”.
4. Lines 108-109, Section 2.2.2: Please, specify which reaction kettle (autoclave) was used for hydrothermal synthesis in order to ensure better reproducibility of your experiments.
A: I've changed the "hydrothermal reaction autoclave" to "hydrothermal reaction reactor" to be more precise.
5. Lines 114-115: Authors said: “The TiO2-Al2O3 composite membrane was placed on the end filtration device (Figure 1(b)) …”. However, I didn’t see a or b marks in Figure 1. Authors should add that or clear the b mark from the text.
A: It is true that the description in the figure and in the text is not consistent, which has been modified and the inappropriate description in the text has been deleted.
6. Lines 149-150: Authors said: “The ZrO2 particle size used in the experiment is mainly distributed in the range of 40 nm to 60nm (Figure 2(c))…”. I think it is incorrect since I saw another dominant size distribution (20-40 nm) from the mentioned figure. Also, remove “used in the experiment” from the text to make sentence more clear.
A: We checked the original photo data of the test and found that there were errors in the description of particle size in the paper. We checked carefully and completed the modification.
7. In my opinion the whole 3.2. Section should be moved to supplementary since it explained reasons for choosing to put intermediate layer (TiO2) between ZrO2 and matrix.
A: If the whole 3.2. Section is moved to supplement, the paper organization will be more concise and clear. However,the experimental process and detection are reflected in the paper, and there is no additional supplementary document. Due to the small space to explain TiO2 as an intermediate layer, 3.2 is placed in the main text for easy understanding.
Lines 163-164: Authors said: “The pore size of the support body ranges from 500 nm to 3500 nm”. From obtained micrographs (Figure 3), I could not see that since it wasn’t measured.
A: The pore size of the purchased alumina support is 1-3 μm, and 500 to 3500 nm is the result of pore statistics for SEM images. It is true that the photo data is not clear enough, and the description is modified.
It would be great if authors should provide BET analyses fore some samples to make some conclusions about specific surface area and porosity.
A: BET test is a good method to analyze the specific surface area and porosity of membrane. However, the membrane in this paper is asymmetric, and the pore diameter and structure are gradual, so BET cannot be directly used for evaluation. In the following experiments, we will peel the nanofiltration membrane, and then use BET to analyze the specific surface area and porosity of nanofiltration membrane.
8. Lines 191-193: “By comparing the morphology of the matrix (Figure 3(c)), it can be seen that the prepared TiO2 intermediate...” I think it is a Figure 3(a).
A: This error has been corrected after inspection, thank you for pointing it out.
9. Since you provided SEM micrographs of intermediate TiO2 layer with and without PVA (Figure 4), you should mention preparation procedures of TiO2 layer with and without PVA in experimental section.
A: The membrane preparation process without adding PVA has been added and shown in the experimental preparation section.
10. Lines 212-213: “Zirconia surface layer was prepared by deposition through terminal filtration device. The microscopic morphology of the membrane is shown in Figure 5”. Figure 5(a)?
A: In order to make it easier to understand, this sentence has been moved to after “the hydrophilicity of membrane and ion filtration [46].” and modified to “Figure 5 shows the microstructure characterization of the composite film and surface XRD phase analysis.”
11. Lines 242-246, Section 3.3.3: The paragraph “The hydrophilicity of ZrO2 prepared with glycerol added was lower than that of TiO2 hydrophilic material. The results show that after the introduction of PEI-modified ZrO2 separation layer, the hydrophilicity was slightly improved, and the contact angle was stabilized at 14.1°. This is mainly due to the large number of hydrophilic groups in PEI, resulting in hydrophilic modification of the separation layer.” It is confusing since water contact angle measurement result was not provided for ZrO2 prepared with glycerol. Also, I did not see the comparison between ZrO2 prepared with glycerol and PEI-modified ZrO2 before (SEM, etc.).
A: In FIG. 6, a set of ZTA surface water contact angles without PEI are added to facilitate a better understanding of the changes in hydrophilicity and hydrophobicity between the surface and the middle layer, and corresponding expressions are also added to the experimental procedures. We focused on exploring the surface of ZTA composite film supplemented with PEI, without contrast scanning electron microscopy. In order to make the results clearer, we will continue to focus on SEM comparison in the future.
12. Lines 298-299, Conclusions: Authors said: “The addition of PEI improved the retention and adsorption ability of the membrane surface”. However, there were not performed experiments which confirms this statement (etc. comparison of retention and adsorption ability of the membrane surface between ZrO2 and PEI-modified ZrO2).
A: Because of the ambiguity caused by the careless expression, we made a little modification. “The addition of PEI improved the retention and adsorption ability of the membrane surface.” has been replaced by “The ZTA composite membrane prepared in laboratory has certain ion interception ability.”
Round 2
Reviewer 2 Report
The manuscript can be published in Coatings.
Reviewer 3 Report
The new version of the manuscript entitled "Preparation of ZrO2/TiO2/Al2O3 nanofiltration lab scale membrane for filtering heavy metal ions" was significantly improved according to reviewers instructions. Authors provided very good response to my review report and took into consideration all my suggestions. So, the paper can be accepted for the publication in the "Coatings" jouranl in presented form.